# Risk Prediction Models for Melanoma: A Systematic Review on the Heterogeneity in Model Development and Validation

**DOI:** 10.3390/ijerph17217919

**Published:** 2020-10-28

**Authors:** Isabelle Kaiser, Annette B. Pfahlberg, Wolfgang Uter, Markus V. Heppt, Marit B. Veierød, Olaf Gefeller

**Affiliations:** 1Department of Medical Informatics, Biometry and Epidemiology, Friedrich Alexander University of Erlangen-Nuremberg, 91054 Erlangen, Germany; isabelle.kaiser@fau.de (I.K.); annette.pfahlberg@fau.de (A.B.P.); wolfgang.uter@fau.de (W.U.); 2Department of Dermatology, University Hospital Erlangen, 91054 Erlangen, Germany; markus.heppt@uk-erlangen.de; 3Oslo Centre for Biostatistics and Epidemiology, Department of Biostatistics, Institute of Basic Medical Sciences, University of Oslo, 0317 Oslo, Norway; m.b.veierod@medisin.uio.no

**Keywords:** melanoma, risk prediction, statistical models, validation

## Abstract

The rising incidence of cutaneous melanoma over the past few decades has prompted substantial efforts to develop risk prediction models identifying people at high risk of developing melanoma to facilitate targeted screening programs. We review these models, regarding study characteristics, differences in risk factor selection and assessment, evaluation, and validation methods. Our systematic literature search revealed 40 studies comprising 46 different risk prediction models eligible for the review. Altogether, 35 different risk factors were part of the models with nevi being the most common one (*n* = 35, 78%); little consistency in other risk factors was observed. Results of an internal validation were reported for less than half of the studies (*n* = 18, 45%), and only 6 performed external validation. In terms of model performance, 29 studies assessed the discriminative ability of their models; other performance measures, e.g., regarding calibration or clinical usefulness, were rarely reported. Due to the substantial heterogeneity in risk factor selection and assessment as well as methodologic aspects of model development, direct comparisons between models are hardly possible. Uniform methodologic standards for the development and validation of risk prediction models for melanoma and reporting standards for the accompanying publications are necessary and need to be obligatory for that reason.

## 1. Introduction

Cutaneous melanoma is one of the most lethal forms of skin cancer and accounts for the majority of skin cancer deaths [1]. Over the past few decades, the incidence of melanoma has risen dramatically worldwide, especially in regions with fair-skinned populations [1,2,3]. While in many European populations, like the UK and the Netherlands, incidence rates increased with estimated annual percentage changes of 4 to 6% event in recent decades, the annual increase in incidence in Australia and New Zealand seems to have leveled off since 1995 [4]. Nonetheless, Australia and New Zealand have the highest incidence rates worldwide, followed by Western Europe, Northern Europe, and the USA [1,3,4]. In 2018, melanoma was the fourth most common cancer in Australia regarding incidence [5]. In Europe (except Southern Europe) and the USA, it ranks fifth and sixth [5].

Melanoma survival is strongly related to tumor thickness at diagnosis and whether the tumor has already metastasized or not [6]. Therefore, early diagnosis is important for the successful treatment of the disease and to keep mortality at a low level [1,2,7].

Melanoma screening programs are essential for early diagnosis [2]. Although population-based screening efforts have the potential to decrease the mortality rate of melanoma, as shown by the SCREEN (Skin Cancer Research to Provide Evidence for Effectiveness of Screening in Northern Germany) project [7,8], screening the whole population is cost-intensive and its efficacy unproven [9,10]. Thus, targeted screening of individuals at higher risk of developing melanoma is regarded as the more effective prevention strategy [11,12]. Targeted screening needs proper identification of the subgroup with a higher melanoma risk. To this end, so called risk prediction models for melanoma have been developed [13]. Based on a combination of risk factors, they assess an individual’s risk of developing melanoma. Tailored risk estimates provided by prediction models have the additional advantage that individuals may interpret this information as more personally relevant for them and may feel encouraged to participate in melanoma screening programs or even favorably change their behavior in order to reduce melanoma risk [14,15].

Due to the growing general interest in cancer risk prediction and the increasing incidence rates of melanoma, a high number of melanoma risk prediction models have been developed over the last few decades. However, the more models are being developed, the more difficult it is for practitioners to judge which model is the most useful [15]. Therefore, details on the performance and reliability of the models, as well as direct comparisons of multiple models, are necessary in order to provide guidance as to which model to use in clinical practice. For melanoma, however, quantitative comparisons of risk prediction models are hitherto limited.

The aim of this systematic review is to give an overview of published risk prediction models for melanoma while highlighting the heterogeneity in definition and assessment of predictors incorporated in these models. Additionally, we describe systematically how these models have been validated and which methods and measures to evaluate their performance have been employed.

## 2. Materials and Methods

### 2.1. Eligibility Criteria and Search Strategy

For our analysis, we searched for studies describing prediction models quantifying the risk of melanoma and identifying people at high risk of developing melanoma, respectively. Only studies using a multivariable prediction model were eligible for this review. The models should provide either absolute risks or risk scores, or report the mutually adjusted relative risks of individual risk factors and risk factor combinations in multifactorial statistical models. As we focus on the prediction of new cases instead of the recurrence of melanoma, we concentrated on models for primary cutaneous melanoma (both invasive and in situ). Furthermore, only studies using a well-defined statistical method for the development of their models and risk predictions derived from these models, respectively, were eligible for this review, while models developed primarily based on expert opinions or consensus meetings were excluded.

We started our search with the systematic reviews of Vuong et al. [15] and Usher-Smith et al. [13], which are the only systematic reviews regarding melanoma risk prediction. They comprise a total of 26 different studies [16,17,18,19,20,21,22,23,24,25,26,27,28,29,30,31,32,33,34,35,36,37,38,39,40,41]. Both systematic reviews were published in 2014 and used similar eligibility criteria. They only included studies developing multivariable prediction models for incident primary cutaneous melanoma providing a measure of relative or absolute risk for members of the general population. Further eligibility criteria of Usher-Smith et al. [13] were that the studies were published as primary research articles and derived the final risk prediction model from their own study data. In contrast, Vuong et al. [15] made no specific restrictions on publication type, study type, and methodology. We adopted the less restrictive eligibility criteria of Vuong et al. [15], but had to exclude two of the 26 studies [38,41], both appearing only in Usher-Smith et al. [13], from further consideration. One study publication [41] is only available in Russian language and could thus not be assessed by us. The other study [38] focused on the identification of risk factors rather than on the prediction of melanoma risk and the identification of people with a high melanoma risk, respectively. The study did not report a prediction model containing mutually adjusted relative risks and thus did not fulfill the eligibility criteria. We performed forward snowballing (searching within the citations to the paper being examined [42]) on these two systematic reviews using Google Scholar and Web of Science. Due to its reported high precision, forward snowballing is a useful approach to support the update of systematic literature reviews [42,43,44], which is why we used this method to find more recent melanoma risk prediction studies. Additionally, we performed an electronic literature search in PubMed from May 1, 2013 (as this is the date the literature search of Vuong et al. [15] ended) up to January 31, 2020. We used combinations of the search terms “risk/risk assessment/probability”, “prediction/model/score” and “melanoma/skin cancer” (see Appendix A for complete search strategy).

### 2.2. Data Extraction

In case of studies developing several models by starting with a base model and adding more risk factors step-by-step, only the full model has been included. Regarding studies with several models but without a differentiation between a base and a full model, the model with the best performance was included. If there was no model with significantly better performance, all models were selected for analysis. For studies with separate models for men and women, or for self-assessment and physician assessment, both models were included separately.

The characteristics of study design, study sample, analytic model, predictive factors included in final models, and outcome measures were abstracted, as well as evaluation and performance measures. Additionally, spatio-temporal information on the location and time of the studies were extracted.

### 2.3. Data Processing

The studies and their prediction models, respectively, were analyzed regarding the following aspects: (1) the spatio-temporal information; (2) the heterogeneity of risk factors in general; (3) the disparities in defining and ascertaining individual risk factors; (4) the validation methods, and (5) the evaluation of model performance.

To provide aggregated information on the geographical location, we allocated the studies to the countries and continents where their data sets originate. Additionally, we performed a temporal synthesis by dividing the entire period of time, in which the studies were published, into eight intervals of four years each.

To report the overall heterogeneity of risk factors, classification of model variables into meaningful groups was performed in order to keep the amount of different risk factors manageable. Variables which have the same meaning but different names (e.g., dysplastic nevi and atypical nevi) were aggregated. In addition, variables that belong to the same topic (e.g., sunburns in childhood, lifetime sunburns and sunburns without further specification) were also combined. One study used a pigmentation score as risk factor that was calculated reproducibly from the variables hair color, eye color, tanning ability and skin color [45]. We therefore applied the four individual factors in our analysis instead of the score. Concerning the variable red hair (RH)-phenotype used in [46], which is a combination of the risk factors hair color, freckles and Fitzpatrick skin type, we proceeded similarly by including the three phenotypic characteristics as separate variables. A full list of all adaptions made is presented in the Appendix A.

To illustrate that the assessment of the same risk factor was not uniform in all studies, the variable nevi was analyzed in detail regarding differences in the size of the nevi that were counted, the body site on which they were counted, the examiner who was counting and the measurement level of the variable. The variable nevi was chosen as it is the most common risk factor in melanoma risk prediction models. Another example is the risk factor sunburns, which we also examined regarding its diverse definitions and measurement levels.

Validation methods comprise internal and external validation. While internal validation relates to the reproducibility of the model and should be performed during model development, external validation refers to the generalizability of the model to other populations [47]. Possible internal validation methods are split sample validation, bootstrap resampling and cross-validation.

Evaluation techniques involve measures of model performance like discrimination and calibration, which are traditional approaches to determine the performance of prediction models. In addition, other more recently suggested performance measures related to the clinical usefulness of a prediction model and reclassification like decision curves and the net reclassification index (NRI) were also considered [48]. Furthermore, we analyzed the difference between older and newer studies regarding both validation and evaluation methods. Therefore, we divided the included studies based on the median year of publication into two equal groups (“studies published up to 2011” and “studies published after 2011”).

The results were summarized using descriptive statistics; frequencies and percentages were gathered and displayed in tabular form. Percentages were relative to the total number of studies or risk prediction models. It must be noted that the total number of studies differs from the total number of risk prediction models, as some studies developed multiple models. In case of subgroup analyses, e.g., when only models including the risk factor nevi are considered, percentages relate to the number of models in this subgroup.

## 3. Results

### 3.1. Study Selection

Altogether, 24 of the 26 studies [16,17,18,19,20,21,22,23,24,25,26,27,28,29,30,31,32,33,34,35,36,37,39,40] from the two systematic reviews [13,15] were included in our analysis. Eight further studies were identified via forward snowballing [46,49,50,51,52,53,54,55], while eight studies were found in PubMed [45,56,57,58,59,60,61,62]. Therefore, we included 40 studies in our analysis (Figure 1).

Thirteen of these studies developed multiple models, up to six models in one study [56]. Thus, the total number of models described in the 40 studies amounted to 66. By only including full models and models that have the best performance, when possible, we reduced the number of risk prediction models to 46.

### 3.2. Study Characteristics

Spatio-temporal information on the location and time of the studies were aggregated and are shown in Figure 2 and Figure 3. Figure 2 displays the distribution of studies according to their continent and country of origin. The majority of studies (*n* = 36) originated from countries under the top 20 with the highest melanoma incidence rates, which include the United States (*n* = 9), Australia (*n* = 8), and Germany (*n* = 4) [63]. Four studies used data sets from several countries with high melanoma incidences for the development of their model.

The temporal distribution of the reviewed studies is shown in Figure 3. The most recent time intervals, 2012–2015 and 2016–2019, show the highest numbers of publications (*n* = 10 each). In fact, half of the studies were published in those eight years, while the remaining 20 studies are spread over the larger time period from 1988 to 2011.

The key information extracted from each study is summarized in Table A1. Study designs used were mainly case-control (*n* = 30) and cohort (*n* = 8). Two studies used published material from meta-analyses to obtain risk estimates. Most of the studies used logistic regression for model development (*n* = 29), but also the Gail method and Cox regression were applied several times (*n* = 5 and *n* = 3 studies, respectively). The Gail method was originally developed to assess the risk of breast cancer and combines risk estimates with incidence and mortality rates [15]. Two studies used the machine learning techniques random forest and decision trees to build their risk prediction models. In three studies, multiple approaches were employed to develop different models. Regarding the reported risk measure, only eight studies calculate the absolute risk for an individual to develop melanoma. A total of 12 studies calculated a risk score, of which 10 defined a cut-off point for being at high risk. Three further studies used relative risks with definitions of high risk. All other studies calculated the relative risk or odds ratio of single predictors or factor combinations without giving a definition of when a person is considered as being at high risk.

The average number of variables incorporated in a model was six (range 1–16). Cho et al. [58] developed their risk prediction model based on only one variable, a genetic risk score (GRS) which comprised aggregated genetic information from 21 single nucleotide polymorphisms (SNPs). All other studies described models including multiple independent variables. One study [55] used data of electronic health records (EHR) and therefore included a large number of variables in the risk prediction model; however, the exact number was not reported. As most of the risk factors were not specified in the publication, this study was excluded in the analysis of predictors included in the models.

### 3.3. Risk Factors Included in the Prediction Models

After grouping all risk factors in categories as described in Appendix A, we obtained 35 different predictors. Besides phenotypic risk factors like nevi, eye color and Fitzpatrick skin type, genetic and demographic risk factors were also used, as well as risk factors related to sun exposure and pigmented lesions (Table 1). The most common predictor is nevi, which has been used in 78% of the risk prediction models, followed by hair color (58%). Polygenic risk scores (PRS) were used in five models. In total, 15 of the 35 risk factors only occur in one or a maximum of two models. Almost all studies (except [58] and [26]) included at least one phenotypic predictor, whereas only one third of the models included genetic risk factors.

Figure 4 displays the frequencies of risk factor combinations as a heatmap. This figure only contains those 20 risk factors that appear in more than two of the risk prediction models. A plot with all 35 risk factors can be found in the Appendix A. The most common predictor combinations are nevi and hair color (*n* = 21, 47%), followed by nevi and freckles (*n* = 15, 33%) and nevi and Fitzpatrick skin type (*n* = 14, 31%).

As studies used different definitions for at first glance identical predictors and employed varying methods of ascertaining them, Table 2 and Table 3 show the heterogeneity in the definition and assessment of risk factors for the two examples, nevi and sunburns. Altogether, 69% of the models using nevi as predictor did not specify the size of nevi that were counted. No uniform procedure was also discernible for the body site on which the nevi were counted. Besides on the entire body (*n* = 16, 47%), nevi were also counted on only one or both arms and/or the back. Additionally, self-assessment of nevus counts by the study participant was approximately as frequent as the assessment by a professional, e.g., nurse or physician. Apart from four models, all others included nevi as a categorical variable. However, each study defined different categories (e.g., none/few/some/many or none/1-2/3-5/6-9/10+). Concerning the risk factor sunburns, six out of 13 studies did not specify how they define sunburns. The most frequent definition was “blistering,” which was used by four studies. Furthermore, five studies asked for sunburns throughout the subject’s lifetime, just as many addressed only sunburns in childhood and three studies did not specify this aspect. The reported measurement level of the variable was either dichotomous (*n* = 8, 61.5%) or categorical (*n* = 5, 38.5%).

### 3.4. Validation and Model Performance

In total, 18 out of all 40 studies used internal data to validate their models and only six used an external data set (Table 4), including three studies that did both internal and external validation. The remaining 19 studies used neither internal nor external validation. One of the studies with external validation used multiple external data sets, while the other five studies used just one external data set. For analyzing the temporal effect on the validation methods, we compared the two subgroups (“studies published up to 2011” and “studies published after 2011”). We found that the proportion of internal validations in the first subgroup is only one quarter compared to more than 60% in the subgroup of more recent studies. Furthermore, 70% of the older studies did not report any validation, while among the studies published after 2011 this proportion decreased to 25%.

Calibration was assessed in 11 studies and discrimination in 29 studies. The area under the curve (AUC) was the most common measure of discriminative performance (*n* = 26 studies, 65%). The proportion of studies published after 2011 reporting the AUC is 90%. Only one study reported measures for the overall model performance. Performance measures related to reclassification and clinical usefulness were less common (*n* = 4 and *n* = 11, respectively). In total, 73% of the studies reported a performance measure. Nine of the 20 studies published up to 2011 did not provide a measure of performance, while in the group of more recent studies, each study reported at least one performance measure.

## 4. Discussion

This systematic review shows that an abundance of melanoma risk prediction models were developed over the last decades, which has already been indicated by the systematic reviews of Vuong et al. [15] and Usher-Smith et al. [13] in 2014. We identified 40 studies reporting 46 risk prediction models that showed substantial heterogeneity in the choice of predictive factors and their definitions. In addition, only little consistency in model evaluation and poor validation were found among the studies.

The substantial heterogeneity of risk factors included in the prediction models is recognizable by the fact that 35 different risk factors were used in the risk prediction models. Whereas only some of them were frequently used, many (*n* = 15) were only used in one or two models. Six of the top 10 most common predictors were the phenotypic factors like nevi, hair color, Fitzpatrick skin type, freckles, skin color, and eye color, with nevi most frequently appearing in three quarters of the models. Thus, phenotypic predictors are currently dominating melanoma risk prediction, which could be traced back to their simple and fast assessment. Genetic factors, whose determination is more complex, are clearly less frequently used. Although polygenic risk scores may have the potential to meaningfully improve the predictive value of risk prediction models, their efficacy in clinical use is unproven [64].

As the heatmap in Figure 4 illustrates, risk factors were combined in various ways across all risk factor groups and only a few factor combinations were frequent. In fact, not even two of all melanoma risk prediction models in our analysis used exactly the same predictors. This heterogeneity of variables may be conditioned by the diversity of countries from where the studies originate. Nevertheless, it complicates the validation by external data sets and direct comparisons between models on the same data set. Existing cohort studies may, for example, not be used for validation due to lack of information on one or more of the variables in the prediction model, and an existing data set used to develop one model does probably not contain all necessary predictors for other models. New data sets explicitly collected for preselected risk prediction models would be necessary.

By means of the two examples, nevi and sunburns, we demonstrated the considerable variation in the definition and assessment of risk factors, which only seem to be identical at first glance, as well as, in many cases, the lack of details in definitions and assessment methods given in the publications. About 70% of the models using nevi as risk factor did not specify the size of nevi counted. English and MacLennan published an IARC protocol for the identification and reporting of pigmented lesions like nevi and atypical nevi in 1990 [65]. Nevertheless, only the publication of Fortes et al. [23] cited the IARC protocol as reference for their nevi assessment, although 28 of the 35 risk prediction models with nevi as risk factor were developed after its publication. Regarding sunburns, a detailed definition enabling the distinction between a mild erythema and a painful sunburn was missing in almost half of the models. However, even if the definition is the same in several models, there are still many differences e.g., in the measurement levels or the categorization of quantitative variables. The more subjective a predictor is, the more variation is seen. This makes the comparability of the risk prediction models even more difficult. In order to externally validate models on independent data sets, uniform methods of data acquisition and a complete reporting of all information are necessary. One recent approach to the problem of non-uniform methods for data collection is given by the MelaNostrum consortium [66], which is a collaboration of researchers and clinicians from Mediterranean countries. They developed a consensus questionnaire of epidemiologic and clinical variables for melanoma risk assessment in order to standardize data collection across different studies, centers and languages [66].

Another significant cause for the heterogeneity of results and variables used in the reviewed studies could be traced back to the use of suboptimal primary data especially when attempting to capture individual UV-exposure over longer periods and behavioral aspects of UV protection. Self-reported sun exposure is commonly used, but it is uncertain how well this correlates with the actual sun exposure [67]. Biases from recalling past sun-related behavior and socially desirable answers are known problems, potentially limiting validity and reliability of exposure assessment [68,69,70]. Even when focusing on sunburn history, a commonly used marker of excessive UV exposure that has originally been believed to be better recalled, the ascertainment in self-administered questionnaires has been shown to lack reproducibility [71] and to underestimate the true extent of sunburns [72]. The validation of questionnaires would provide evidence for the magnitude of these problems, but this kind of validation is rare [67,68,73].

The analysis of the evaluation methods employed in the studies showed that the actual standard of validation and reporting of performance measures is quite poor. Nearly half of the studies (*n* = 18) validated their models internally using methods like bootstrapping and cross-validation, but only six out of 40 studies externally validated their models. The direct comparison of older and more recent studies reveals, however, an encouraging temporal effect. The proportion of studies without validation is clearly lower in the subgroup of studies published after 2011 compared to the subgroup of studies published up to 2011 (25% versus 70%). Nevertheless, there are still several exceptions among the newer studies [16,39,46,53,58], which did not report to have performed any validation of their model. Internal validation is useful for ensuring reproducibility of the model and stability of the predictor selection, as well as avoiding overfitting [47,74]. Applied to the data set used for developing the model, prediction models often perform substantially better than on external data sets, which can lead to overestimation of the predictive ability [75,76]. However, the models’ performance on independent samples is a particularly valuable indicator for their discriminatory ability when applied in clinical practice [76]. Models developed for one population may not be valid in another population, as specific predictors do not necessarily apply in various regions. Therefore it is important to externally validate risk prediction models in order to verify generalizability and transportability of the model to other cohorts [47]. To reduce overoptimistic expectations of model performance on independent data, Steyerberg and Harrell [77] proposed internal-external validation procedures during model development.

Considering the increasing number of different prediction models, studies about the external validation of multiple models may be the best way for a direct comparison of the models and to determine which model performs best on an independent data set [75]. Due to missing information about the model development and risk factor assessment in many cases, as discussed in previous sections, external validations particularly by independent researchers are rare [74]. One positive exception constitutes the study of Vuong et al. [49], who externally validated their risk prediction model using four independent data sets from other studies.

Further discrepancies among the studies were found concerning the performance measures. More than one-quarter of the publications (*n* = 11) did not report any performance measures, although a quantification of model performance is essential to evaluate if the model is suitable for application [48]. However, we observed an encouraging temporal effect concerning the evaluation of model performance too. For all categories of performance measures the proportion of studies reporting this kind of performance measure is higher in the subgroup of more recent studies than in the subgroup of older ones. Widely used statistical concepts for the evaluation of risk prediction models and the quality of prediction are discrimination and calibration. For the identification of individuals with a high risk of developing a certain disease, the discriminative ability is the most important property of the prediction models [78]. Almost 75% of the studies reported measures for the discriminative ability e.g., the AUC or c-index. Good calibration and discrimination are necessary but not sufficient for clinical usefulness. Hence, measures regarding the clinical usefulness can be regarded as essential to determine whether the model is beneficial for clinical practice and decision making [75,78]. Such performance measures related to clinical usefulness only emerged in the past few years, which may be the reason for their sparsity among the risk prediction models included in our analysis.

Even though this review relates to risk prediction models for melanoma, the problems identified are not limited to this area of medicine, as demonstrated by other publications of multivariable prediction or prognostic models addressing cardiovascular diseases, colorectal cancer, and diabetes type 2 [79,80,81,82,83]. Their results confirm the lack of validation and the need for more uniform assessment methods and independent validation. Due to the current focus on risk prediction models in general, it is an interdisciplinary problem that significantly more models have newly been developed than existing models were validated [74,75]. Furthermore, the fact that many risk factors are only used in one or two models indicates that the primary focus of most studies has been the identification of new predictors and the development of new risk prediction models, instead of the improvement and validation of existing models [79].

Complete reporting of how a prediction model has been developed and validated is necessary to objectively appraise the usefulness of the model [83]. Therefore it is a general requirement for all studies developing risk prediction models to report key details on model performance and their development process. The “Transparent Reporting of a multivariable prediction model for Individual Prognosis Or Diagnosis” (TRIPOD) statement [84], published in 2015, provides a checklist of 22 items essential for transparent reporting of a prediction model study. We found that only three of 10 studies published after the TRIPOD statement, reported their methods and results according to the statement and cited it as reference.

Considering the insufficient evaluation and the lack of quantitative comparisons, it is not surprising that in clinical practice no risk prediction model has been fully established in its entirety. Instead the individual risk is still mostly estimated based on single risk factors like count of nevi (>100 melanocytic nevi), congenital nevi (>40 cm in diameter) and Fitzpatrick skin type [85]. However, not all risk prediction models have been developed for clinical practice, e.g., the studies Barbini et al. [17], Whiteman and Green [35] and Fang et al. [59] are preliminary studies. Other recent examples are the melanoma risk prediction models of Olsen et al. [52] and Vuong et al. [49], which are intended for community use in Australia as online risk calculators [86,87].

There are some limitations of our work that require consideration. First of all, the identification of studies reviewed here did not follow the PRISMA standard [88] and thus may not have the same degree of systematicity and objectivity. We performed, however, a comprehensive and careful, well-documented search for melanoma risk prediction models and identified several studies published after the systematic reviews of Vuong [15] and Usher-Smith [13] which were the starting points of our search. In fact, we found 16 publications of new melanoma risk prediction models that were published in the past six years. This relatively high number confirms the topicality of melanoma and the global interest in its prevention. A further limitation relates to the missing comparison of discriminative ability and overall performance of the different prediction models. We intentionally refrained from addressing this topic due to its methodologic complexity, which cannot be solved based on the information given in the study publications. Due to our abandonment of comparing the models with respect to their properties in practical applications, we also dropped the assessment of risk of bias for the studies having developed the prediction models. Shortcomings in study design, conduct and analysis can cast doubts on the study results and should be taken into consideration when evaluating prediction models comparatively in systematic reviews. Recently, the PROBAST tool has been developed through a consensus process to assess risk of bias and applicability of prediction model studies [89]. For our review we avoided using the PROBAST tool since its intention was beyond the scope of our project.

## 5. Conclusions

In summary, we found substantial heterogeneity in several important aspects of published risk prediction models for melanoma. Although a large number of models has been published, external validation is largely missing and direct comparisons between models are hardly possible. Consequently, there is no consensus in sight how to predict individual melanoma risk appropriately. Uniform standards for the assessment and documentation of predictors, as well as better adherence to reporting guidelines like TRIPOD are necessary and need to be obligatory in order to ultimately achieve a convincing solution of this problem.

## Figures and Tables

**Figure 1 ijerph-17-07919-f001:**
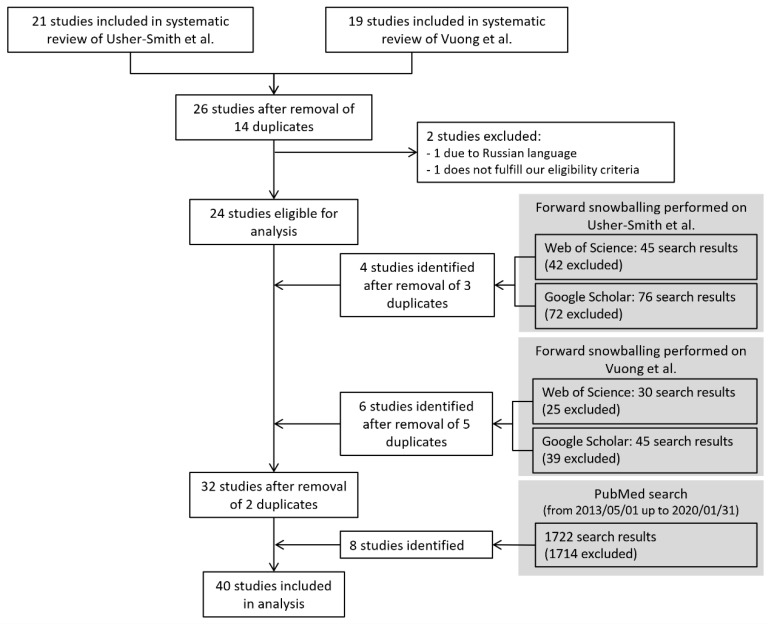
Flow diagram for the identification of studies developing risk prediction models for melanoma.

**Figure 2 ijerph-17-07919-f002:**
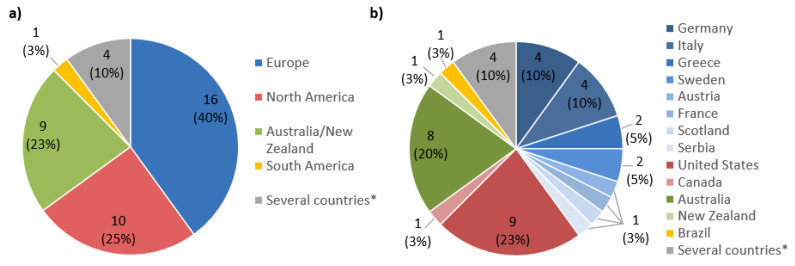
Geographical location of studies. (**a**) Distribution of studies according to the continents of their origin. (**b**) Distribution of studies according to their country of origin. (*n* = 40 studies). * Four studies used data sets from multiple countries with high melanoma incidences for the development of their risk prediction model.

**Figure 3 ijerph-17-07919-f003:**
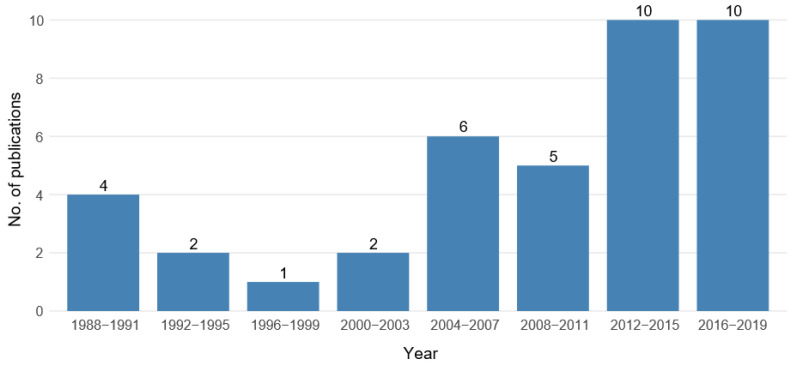
Temporal distribution of the reviewed studies showing the number of publications in eight time intervals of four years each. (*n* = 40 studies).

**Figure 4 ijerph-17-07919-f004:**
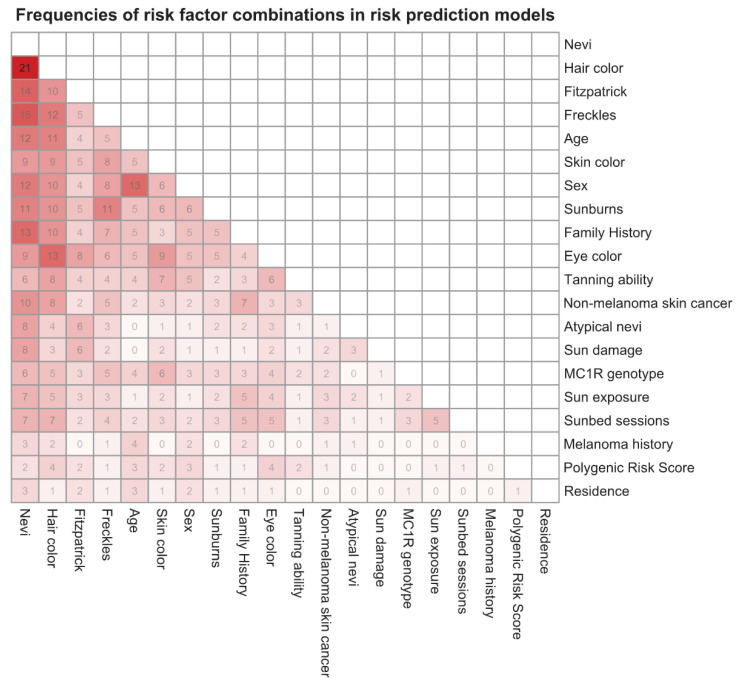
Heatmap indicating joint occurrences of risk factor pairs in risk prediction models for melanoma. Only risk factors occurring in more than two risk prediction models are included. Each number represents the absolute frequency of the corresponding risk factor combination. The darker the field the more frequent is the corresponding risk factor combination (*n* = 45 models). MC1R = melanocortin 1 receptor.

**Table 1 ijerph-17-07919-t001:** Absolute (*n*) and relative frequencies (%) of predictive factors included in the risk prediction models for melanoma (*n* = 45 models *).

Risk Factors	*n*	%
**Phenotypic factors**		
Nevi	35	77.8
Hair color	26	57.8
Fitzpatrick	17	37.8
Freckles	16	35.6
Skin color	15	33.3
Eye color	14	31.1
Tanning ability	10	22.2
**Genetic factors**		
MC1R genotype	7	15.6
Polygenic risk score	5	11.1
SNPs	1	2.2
**Demographic factors**		
Age	16	35.6
Sex	15	33.3
Family history of melanoma	13	28.9
Residence	3	6.7
Level of education	1	2.2
Country of birth	1	2.2
Health insurance	1	2.2
Ethnicity	1	2.2
1st degree relative with large or unusual moles	1	2.2
**Sun exposure**		
Sunburns	13	28.9
Sunbed sessions	7	15.6
Sun exposure	7	15.6
Occupational sun exposure	2	4.4
Use of sunscreen	2	4.4
**Skin lesions**		
Non-melanoma skin cancer	10	22.2
Atypical nevi	10	22.2
Sun damage	8	17.8
Melanoma history	5	11.1
Congenital nevi	2	4.4
Previous skin lesions treated destructively	2	4.4
Suspicious melanocytic lesions	1	2.2
Changing moles	1	2.2
**Other risk factors**		
Skin checks	2	4.4
Hormonal contraceptive therapy	1	2.2
Age on arrival in Australia	1	2.2

Abbreviations: MC1R = melanocortin 1 receptor, SNP = Single Nucleotide Polymorphism. * Study of Richter et al. [55] excluded due to limited reporting of predictors.

**Table 2 ijerph-17-07919-t002:** Heterogeneity of the risk factor nevi in risk prediction models for melanoma regarding four aspects: minimum size of nevi to be counted, body area of nevi count, type of nevi assessment, and measurement level (*n* = 35 models).

Risk Factors	*n*	%
**Size of nevi**		
≥2 mm	7	20.0
≥5 mm	2	5.7
>3 mm	1	2.9
≥2 mm and ≥5 mm, respectively	1	2.9
Not reported	24	68.6
**Site of nevi count ^(1)^**		
Entire body	17	48.5
Both arms	6	17.1
Right arm	2	5.7
Forearm and back	2	5.7
Back	2	5.7
Left arm	1	2.9
Not reported	6	17.1
**Assessment**		
Physician/nurse/trained examiner	15	42.9
Self-reported	13	37.1
Not reported	7	20.0
**Measurement level**		
Categorical	31	88.6
Metric	2	5.7
Dichotomous	1	2.9
Not reported	1	2.9

^(1)^ One model with nevi counted on two different sites.

**Table 3 ijerph-17-07919-t003:** Heterogeneity of the risk factor sunburns in risk prediction models for melanoma regarding three aspects: definition of sunburn, time period of sunburn occurrence and measurement level (*n* = 13 models).

Category	*n*	%
**Definition of sunburn**		
Blistering	4	30.8
Pain and erythema or blisters for >24 h	1	7.7
Painful	1	7.7
Peeling of skin	1	7.7
No explanation given	6	46.2
**Time period**		
Childhood	5	38.5
Lifetime	5	38.5
Not reported	3	23.1
**Measurement level**		
Dichotomous	8	61.5
Categorial	5	38.5

**Table 4 ijerph-17-07919-t004:** Absolute (*n*) and relative frequencies (%) of methods used when evaluating risk prediction models for melanoma regarding the methodological type of validation and the type of measures describing model performance (*n* = 40 studies) *.

	Studies Published up to 2011 (*n* = 20)		Studies Published after 2011 (*n* = 20)		All Studies (*n* = 40)	
**Validation**	** *n* **	**%**	** *n* **	**%**	** *n* **	**%**
**Internal validation**	**5**	**25.0**	**13**	**65.0**	**18**	**45.0**
Cross-validation	1	5.0	5	25.0	6	15.0
Split sample	3	15.0	3	15.0	6	15.0
Bootstrapping	1	5.0	5	25.0	6	15.0
**External validation**	**1**	**5.0**	**5**	**25.0**	**6**	**15.0**
Both internal and external validation	0	0.0	3	15.0	3	7.5
Neither internal nor external validation	14	70.0	5	25.0	19	47.5
**Performance measures**	** *n* **	**%**	** *n* **	**%**	** *n* **	**%**
**Calibration ^(1)^**	**2**	**10.0**	**9**	**45.0**	**11**	**27.5**
Hosmer–Lemeshow test	2	10.0	7	35.0	9	22.5
Graph (plot/intercept/slope)	0	0.0	3	15.0	3	7.5
Calibration in the large	0	0.0	1	5.0	1	2.5
**Discrimination ^(2)^**	**9**	**45.0**	**20**	**100.0**	**29**	**72.5**
AUC	8	40.0	18	90.0	26	65.0
C-index	0	0.0	3	15.0	3	7.5
Discrimination slope	0	0.0	1	5.0	1	2.5
ROC plot (without AUC calculation)	1	5.0	0	0.0	1	2.5
**Overall model performance ^(3)^**	**0**	**0.0**	**1**	**5.0**	**1**	**2.5**
Brier score	0	0	1	5.0	1	2.5
Nagelkerk’s R^2^	0	0	1	5.0	1	2.5
**Reclassification ^(4)^**	**0**	**0.0**	**4**	**20.0**	**4**	**10.0**
Net reclassification improvement	0	0.0	4.0	20.0	4	10.0
Integrated discrimination index	0	0.0	2.0	10.0	2	5.0
**Clinically usefulness**	**3**	**15.0**	**8**	**40.0**	**11**	**27.5**
Sensitivity/specificity	3	15.0	5	25.0	8	20.0
Decision curve	0	0.0	3	15.0	3	7.5
**No performance measure at all**	**11**	**55.0**	**0**	**0.0**	**11**	**27.5**

* For extended table with all references see Appendix A. ^(1)^ Two studies reported multiple calibration measures. ^(2)^ Two studies reported multiple discrimination measures. ^(3)^ One study reported both performance measures. ^(4)^ Two studies reported both reclassification measures.

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
