# Peer review of "Risk Prediction Models for Melanoma: A Systematic Review on the Heterogeneity in Model Development and Validation"

_ijerph, 2020, doi:10.3390/ijerph17217919_

Round 1

Reviewer 1 Report

The manuscript by Kaiser et al. is a very important contribution to the field.

This study concerns better data and could have been inspired by Altman BMJ 94.

Many studies executed are similar to previous research but not necessarily better.

A subject that remains unexplored in this manuscript may contribute a significant part of the origin of the heterogeneity of results and variables used in the reviewed studies: Sub optimal (poor) primary data in terms of uv-exposure and strategies and measures of avoidance of exposure. Studies concerning more precise data have been carried out by groups of at least Bispebjerg Hospital, Queensland University in addition to my own. It is unfortunate that studies are choosing convenient data collection over more precise data. Also not handled by melanostrum. One should be lucky to build a stable house on an unstable foundation.

I suggest adding a paragraph about this in the discussion.

Minor suggestions:

Title: Identify as systematic review in title.

Introduction:

  1. Line states ‘CM is one of the most lethal forms of skin cancer…’. I suggest to rephrase as I don’t know any other.
  2. Sentence: What is a dramatic rise? I suggest e.g. a comparison to other cancers or an annual rate change to give an idea of the level of rise.

Results:

Line 166-67 wording

Reviewer 2 Report

The authors reviewed multiple risk prediction models for melanoma from published references. They found that direct comparisons between models are very difficult  due to heterogeneity in model  development as well as validation.  Therefore, they concluded that uniform methodologic standards for the  development and validation of risk prediction models for melanoma are required. This work is very interesting to the field and well written. Only one minor issue needs to be fixed: there are two Figure 1. 

Reviewer 3 Report

Despite the heterogeneity of the analyzed studies, it would have been interesting whether, on a nominal scale, there could be a set of more or less appropriate risk factors for melanoma.

Reviewer 4 Report

The authors of this study report a comprehensive overview of risk prediction models for cutaneous melanoma. Building upon two previous papers on the same topic, they extended the pool of published studies by individuating some additional works from PubMed and employing the forward snowballing method.  

First and foremost, the authors should be complimented for undertaking such a challenging endeavour in what seems to be a highly intricated field. Overall, they managed to depict a grim picture of the current state of the art in the area of risk assessment for melanoma. The clear impression is that previous investigations from a range of research teams, Mediterranean consortia, and other groups have been moving along heterogeneous pathways and with no coordination.

Moreover and remarkably, from a methodological standpoint, the authors demonstrated that the current searching tools are far from being reliable. This was shown by the fact that there was any overlapping between the papers retrieved using PubMed (n=8) and snowballing (n=10), which is quite disconcerting.   

Therefore, I believe their contribution is welcome, although some changes are needed. These could help to improve the quality of the paper and allow the reader to evaluate the mountain of such hard-to-interpret literature.

In my opinion, the authors intended to provide a qualitative analysis of state of the art in the field of risk prediction models. In a certain way, they found and reported a long list of critical issues (limited consensus on risk factors, their definition and assessment; heterogeneity of approaches; missing information; lack of validation and performance measures; incomplete reporting).

However, this information was already available 4-5 years ago. Therefore, if they want to escape from this “mare magnum” and stimulate the scientific community to move in a more coherent direction, I suggest analysing the retrieved literature according to the shared quality criteria that the authors themselves correctly mentioned in their Discussion (e.g. TRIPOD). The adherence (or not) to available reporting guideline could be an important piece of information at this stage.

Along with this, it is highly recommended that authors focus on the most relevant findings and synthesise results as much as possible (this applies also for the analyses alredy performed).    

Round 2

Reviewer 3 Report

The revised manuscript has addressed most of the comments of the reviewers.

The work is of significant interest for epidemiologists as it delivers a critical appraisal of used methods in risk assessment of cancer predisposition.